# A New Isolated Fungus and Its Pathogenicity for *Apis mellifera* Brood in China

**DOI:** 10.3390/microorganisms12020313

**Published:** 2024-02-01

**Authors:** Tessema Aynalem, Lifeng Meng, Awraris Getachew, Jiangli Wu, Huimin Yu, Jing Tan, Nannan Li, Shufa Xu

**Affiliations:** 1State Key Laboratory of Resource Insects, Institute of Apicultural Research, Chinese Academy of Agricultural Sciences, Beijing 100093, China; tessema70@yahoo.com (T.A.); menglifeng@caas.cn (L.M.); awraris2007@yahoo.com (A.G.); wujiangli@caas.cn (J.W.); yuhuimin2018@163.com (H.Y.); 17384769626@139.com (J.T.); linan_caas@163.com (N.L.); 2College of Agriculture and Environmental Science, Bahir Dar University, Bahir Dar P.O. Box 26, Ethiopia

**Keywords:** honeybee, larvae, fungi, *Rhizopus oryzae*, pathogenicity

## Abstract

In this article, we report the pathogenicity of a new strain of fungus, *Rhizopus oryzae* to honeybee larvae, isolated from the chalkbrood-diseased mummies of honeybee larvae and pupae collected from apiaries in China. Based on morphological observation and internal transcribed spacer (ITS) region analyses, the isolated pathogenic fungus was identified as *R. oryzae*. Koch’s postulates were performed to determine the cause-and-effect pathogenicity of this isolate fungus. The in vitro pathogenicity of this virulent fungus in honeybees was tested by artificially inoculating worker larvae in the lab. The pathogenicity of this new fungus for honeybee larvae was both conidial-concentration and exposure-time dependent; its highly infectious and virulent effect against the larvae was observed at 1 × 10^5^ conidia/larva in vitro after 96 h of challenge. Using probit regression analysis, the LT_50_ value against the larvae was 26.8 h at a conidial concentration of 1 × 10^5^ conidia/larva, and the LC_50_ was 6.2 × 10^3^ conidia/larva. These results indicate that the new isolate of *R. oryzae* has considerable pathogenicity in honeybee larvae. Additionally, this report suggests that pathogenic phytofungi may harm their associated pollinators. We recommend further research to quantify the levels, mechanisms, and pathways of the pathogenicity of this novel isolated pathogen for honeybee larvae at the colony level.

## 1. Introduction

Honeybees are globally important to food production, as they facilitate the pollination of about 80% of all insect-pollinated crops and the conservation of flowering plant biodiversity [1,2,3,4]. Honeybees explore their ecological zones to collect honey, pollen, propolis, water, and other essentials they need for survival [5,6,7]. During this collection process, honeybees inevitably interact with either beneficial or harmful substances in the ecosystem, such as microbes, parasites, and fungal spores, which they then carry to the hive [8,9]. Honeybees live in large social colonies, with thousands of individuals living together; they exhibit trophallaxis and nursing behaviors [10,11], which makes it easy for microbes to spread in honeybee colonies. Additionally, honeybee products can also be contaminated with foreign substances the honeybees bring in, such as dust and airborne microbes [12,13]. Therefore, it is particularly important to evaluate the safety of honeybee collection processes.

Honeybees interact with fungi in a wide range of ways, from parasitic to symbiotic. The majority of both beneficial and harmful microbial communities associated with honeybees are acquired from foraging activity [14,15]. Although most of the collected fungal spores do not establish themselves on the honeybee or within the beehive, some fungal species are pathogenic to honeybees [16]. The well-studied honeybee fungi are those belonging to the genus *Ascosphaera*, the causal agent of honeybee chalkbrood disease [17,18]. Another extensively documented bee-associated fungus is in the genus *Aspergillus*, which parasites both honeybee adults and larvae, producing mycotoxins that are toxic to honeybees [19]. Nosemosis is a common bee disease that is closely related to fungi in the genus *Vairimorpha* [20]. Others, such as *Rhizopus* spp. and *Mucor hiemalis*, are considered to be pathogenic to honeybees in certain stress conditions [21]. Some fungal species are saprophytic on honeybee products and combs, and their pathogenic spores are harmful to both brood and adult honeybees [22], predisposing them to further attacks from parasites and predators, and thus hampering productive beekeeping [23,24]. Physical, chemical, and biological stressors may influence the development and difficulty of treatment of entomo-pathogenic fungal infections in honeybees, including mycelial growth that may lead to the death of larvae due to mechanical and enzymatic tissue damage [25,26].

Recently, a number of studies have shown that fungi have extensive roles in bee behavior, development, survival, and fitness [27]. However, less research has been conducted on honeybees. Flower-associated yeasts may serve as an olfactory cue to attract bee foraging, but this does not seem to work on honeybees or may even repel the honeybee [28,29]. Honeybees have occasionally been observed to collect fungal spores in the genera *Melampsora*, *Uromyces*, *Zaghouania*, and *Podosphaera* when floral resources are scarce for unknown reasons, these spores may aid in pollen nutrition and preservation, which is beneficial for honeybee health [30,31]. Fungal cells and their metabolites may act directly as food sources or nutritious supplements, such as amino acids, sterols, and vitamins [32,33]. Fungi can reduce the pathogen load of honeybees by competing with pathogens for growth or may enhance bee immunity [34,35]. There is also a case report that yeast diets can reduce *Nosema* infection in honeybees [36].

*Rhizopus* (Zygomycetes) is a cosmopolitan, ubiquitous, mass-spore-forming genus that includes entomopathogenic fungi [37]. It includes species such as *R. oryzae*, *R. oligosporus*, *R. circinans*, *R. arrhizus*, *R. delemar*, *R. microspores*, and *R. stolonifera* [38]. *Rhizopus* is fast-growing and their infection of fruits and other types of food is characterized by white mycelia and black sporangiospores during the decay process; another principal characteristic is the formation of rhizoids [39,40]. Some *Rhizopus* species, such as *R. oryzae* and *R. stolonifera*, are weak parasites of ripening honeybee-pollinated fruit crops, including apple, peach, strawberry, citrus, persimmon, pear, and pumpkin [39,41,42,43,44]. Following the bees’ foraging activities on infected crops, fungal spores gain entry and become established in the beehive by dissemination through direct contact and food contamination [45,46]. *Rhizopus* spp. has been detected in honeybee bread, crop, midgut, and prepupae [47,48]. Some studies have defined it as a pollen provision contamination; it can spoil bee bread in the right conditions, resulting in a shortage of food supply and a population decline [49], while other studies considered it a natural defense against the pathogenic fungal disease of chalkbrood via microbe-microbe competition or by producing growth-inhibitory substances, but this is only speculation [50,51]. *Rhizopus* was found to be a primary cause of provision spoilage for some soil-nesting bees; *Rhizopus stolonifer* was isolated from the dead brood of the alkali bee and was considered to be the cause of death [48]. Thus, the aims of this study were to (1) observe the occurrence of and isolate *R. oryzae* in cadavers taken from managed honeybee colonies in China and (2) analyze in vitro infection and its pathogenicity for honeybee brood.

## 2. Materials and Methods

### 2.1. Isolation, Morphological Characterization, and Culture Conditions of R. oryzae

Between April and May 2018, brood mummy samples were collected from outside of colonies from four beekeeping sites in China (one in Henan province, one in Jiangsu province, and two in Beijing, where honeybee brood deaths had been recorded. A sample surface was sterilized using 70% ethanol for 1 min and then washed with de-ionized water three times. The brood mummies were then cut into smaller sizes and cultured on potato dextrose agar (PDA) Petri dishes, with a diameter of 9 cm, containing 30 mg/L streptomycin to suppress bacterial growth. The Petri dishes were placed in an incubator at 30 °C for 3 d. Fungal growth was observed using a light microscope, prior to strain selection and isolation. Slides containing hyphal parts were also prepared and observed using a light microscope. To purify the fungus, we collected samples from the isolates using the single-spore isolation method [52] with slight modifications. After purification, we observed the growing hyphal morphology, sporangiospores, sporangia, and sporangiophores of each single fungal mycelium from the inoculated plates, to characterize the fungal hyphal growth, conidia, texture, and spore structures. We collected pure spores from the PDA Petri dishes to grow the mycelia in potato dextrose broth (PDB) (Biomed Co., Beijing, China) for DNA extraction and molecular identification. The experiments were conducted in triplicate and samples were stored at −80 °C prior to analysis.

### 2.2. DNA Extraction and PCR Amplification

Mycelia from the fungal isolates and in vitro-infected larvae were prepared for DNA extraction by culturing spores in PDB, which were then placed on a reciprocal shaker at 140 rpm and incubated at 30 °C for 6 d. A Qiagen fungal DNA extraction kit was used for fungi DNA extraction. The concentration and other quality indicator values of the DNA were measured using the NanoDrop 2000 spectrophotometer (Thermo Fisher Scientific Inc., Waltham, MA, USA). Once high-quality DNA had been extracted from the fungal isolates, PCR was carried out to amplify the complete ITS sequences, using 40 ng/µL of DNA with a forward primer (5’-TCCGTAGGTGAACCTGCGG-3’) and reverse primer (5’-TCCTCCGCTTATTGATATGC-3’). Then, 25 μL of Taq polymerase-based reaction mix was employed in a PCR reaction under the following conditions: initial denaturation at 94 °C for 3 min, followed by 30 cycles of denaturation at 94 °C for 30 s, 59 °C for 30 s, and 72 °C for 30 s, with a final extension at 72 °C for 5 min and rapid cooling to 4 °C. Electrophoresis was used to check the quality of the PCR products, using 4 μL of PCR reaction product on 2% (*w*/*v*) agarose gel. The resulting PCR products (601 bp) were sent to the Biomedical Molecular Research Laboratory, Beijing, China for determination of the complete ITS sequences; these sequences were analyzed using the NCBI nucleotide BLAST tool. The experiments were replicated three times and samples were stored at −80 °C until analysis.

### 2.3. Phylogenetic Analysis

To determine the isolated fungus species’ identity and the infrageneric relationships in the genus, phylogenetic analysis was carried out for the highly identified sequences of these fungi. Multiple sequences collected from NCBI were aligned using MAFFT (version 7.453). Gblocks (version 0.91b) was used to filter poorly aligned positions and divergent regions. A phylogenetic tree of the isolated fungus, based on ITS region, was constructed using the IQ-tree program (version 1.6.12), using the ultrafast bootstrap parameters and suggested models. The reliability of the tree was measured with bootstrapping using 1000 replicates. An outgroup in the analysis was the fungus *Mucor indicus* ITS sequence.

### 2.4. Bioassay of Pathogenicity

We tested the pathogenicity of the isolated fungal strain in vitro using spores collected in the laboratory, following the method described by Wubie [53] with slight modifications. In brief, oral inoculation of in vitro-reared second-instar honeybee worker larvae was used to investigate the differences in the pathogenicity of *R. oryzae*. Here, 1-day-old honeybee worker larvae were grafted to 24-well individual larva-rearing cells. Royal jelly-based larval food [54,55] was prepared, comprising 50% fresh royal jelly mixed with a solute of 6% D-fructose (*w*/*v*), 6% D-glucose (*w*/*v*), and 1% yeast extract (*w*/*v*) in 37% of sterile deionized water, which was pre-warmed to 34 °C. The larvae were fed on this diet once daily and were reared in an incubator at 34 ± 0.5 °C with 70% relative humidity.

To assess the isolate’s pathogenicity, we conducted a challenge inoculation experiment, in which about 1 × 10^2^, 1 × 10^3^, 1 × 10^4^, 1 × 10^5^, and 1 × 10^6^ *R. oryzae* spores were mixed with 5 µL of larval diet and orally administered to the honeybee worker larvae. The infected larvae, showing abnormal dark grey skin coloration and back apical midrib melanization, were observed daily and the mortality rates were recorded. The number of hours post inoculation (hpi) (0 h, 24 h, 48 h, 72 h, 96 h, and 144 h) was set to determine whether the fungal infection depended on time. Koch’s postulates [56] were applied by using healthy second-instar honeybee worker larvae, which were milky white and shiny, to establish that *R. oryzae* caused diseases in the honeybee brood. After the infection experiment, identification of the fungus at the DNA level was carried out, following the protocol described above.

### 2.5. Statistical Analysis

All the bioassay data were analyzed using the SPSS (version 20.0 SPSS Inc., Chicago, IL, USA) statistical software package. To observe the effects of conidial concentration and exposure time and their interaction with the mortality of larvae, a repeated-measures ANOVA was performed. The mean difference of mortality was compared using Tukey’s multiple range test, where *p* < 0.05 was used as an indicator of statistical significance. The median lethal concentration (LC_50_) and time (LT_50_) values were calculated using Probit regression analysis.

## 3. Results

### 3.1. Isolation and Morphological Characterization of the New Fungal Isolate

The brood mummy samples were collected from the apiary where larval disease had occurred and where samples showed symptoms of honeybee chalkbrood disease. Among the fungi isolated from the brood mummy samples, one was different from the previously isolated *Ascosphaera apis*. This fungal colony grew aggressively on PDA within 24 h and was preliminarily identified as *R. oryzae*, based on the morphological profile and conidia analysis (Figure 1A,B) [57].

The colonies growing on PDA were initially white and cottony, then they became heavily speckled with sporangia, and finally became brown-grey to black-grey (Figure 2A–C). They spread rapidly, with stolons fixed to the substrate by rhizoids at various points (Figure 2D). Following incubation for 48 h at 30 °C, the mycelial height was 10–15 mm, touching the Petri dish cover plates (Figure 2B). The sporangiophores were straight, smooth-walled, simple, or branched, non-septate, and long and arose from the stolon opposite rhizoids, usually in groups (Figure 2E). The sporangia were globose, almost transparent at first, before turning black with many spores (Figure 2E). The sporangiospores were unequal, numerous, irregular, oval, and angular, with striations (Figure 2F). The rhizoids and stolons were dark brown (Figure 2G).

### 3.2. Internal Transcribed Spacer (ITS) Sequence Analysis

PCR product quality (Figure 3A,B) and amplified ITS DNA complete sequence alignment confirmed that the isolated fungus was *R. oryzae*, with 99.83% nucleotide BLAST similarity (Figure 4A). Further phylogenetic analysis showed that the isolated fungus in this analysis was classified into the *R. oryzae* clade with high bootstrap support (100%), indicating that the isolated strain was *R. oryzae* (Figure 4B). Taken together, based on multiple sequence alignments and phylogenetic analysis of the ITS region, the results show that the isolated fungus in this study can be considered a new pathogenic isolate of the *R. oryzae* involved in *A. mellifera*.

### 3.3. Pathogenicity Evaluation

Different concentrations of fungal spores caused significant differences in the death rate of honeybee larvae (Appendix A). The mortality of infected honeybee worker larvae significantly occurred within 24 h post-inoculation at spore concentrations of 1 × 10^5^ and 1 × 10^6^ (Appendix A). This was characterized by a change in larval skin color from dark brown to black. Most orally spore-inoculated larvae died and exhibited dehydration, abnormal dark grey skin color, rigidity, and back apical midrib melanization (Figure 5). After 144 h, the surfaces of the dead larvae were extensively covered with conidia and mycelium (Figure 6). Conidia inoculated larvae showed slower growth rates, darker brown skin color, higher mortality rates, and greater mycelial growth, which was preceded by the production of dense yellow sporangiospores, than the control larvae (Figure 6). The disease symptoms on larvae following inoculation with the *R. oryzae* isolate were similar to those seen in the honeybee larvae mummies collected from the apiary sites.

There were significant differences in the in vitro pathogenicity of *R. oryzae* to honeybee worker larvae between the experimental and control groups across the developmental stages (*p* < 0.05). Repeated-measures ANOVA analysis revealed that the mean percentage mortality of *R. oryzae* was conidial-concentration- and time-dependent (Appendix A). The mortality rate increased with an increase in conidial concentrations and incubating time (Figure 7A,B). The mortality increased rapidly from 24 h to 72 h at conidial concentrations of 1 × 10^5^ and 1 × 10^6^, and the maximum mortality (98%) was recorded at 96 h (Figure 7A).

Probit regression analysis was performed to assess the virulence of *R. oryzae* for honeybee larvae. The median lethal concentration (LC_50_) value was 6.2 × 10^3^. The median lethal time (LT_50_) value was 25.50 h at 1 × 10^6^ conidia per larvae and 26.8 h at 1 × 10^5^ conidia per larvae (Appendix A).

## 4. Discussion

Recent studies have found that the cross-infection of host species by pathogens occurs more frequently than previously reported [58,59]. *R. oryzae* is common in the natural environment. Honeybees inevitably carry back this fungus to their hive when they collect nectar and pollen [60]. Although the infection of honeybee larvae by *Rhizopus* spp. is not common, this study successfully isolated and demonstrated the potential pathogenicity of this new *R. oryzae* strain from honeybee mummies to larvae, as a brood pathogen. The evaluation of the pathogenicity and virulence of this new fungus showed that it had high pathogenicity for honeybee larvae in laboratory cultures. Thus, the spread of this new strain to honeybee colonies could have deleterious effects on their individual health and pollinating functions. This study is the first report published on the pathogenicity of *R. oryzae* in honeybees, which suggests a potential risk to the health of honeybee colonies and crop pollination safety.

### 4.1. R. oryzae, a New Isolated Fungus That Infects Honeybee Larvae in Artificial Culture

With the development of biotechnology, morphological identification using light microscopy and electron microscopy, combined with molecular identification, has made fungal identification more accurate. Based on morphological and molecular analyses, the isolated fungus in this study was placed within a clade comprising *R. oryzae* reference isolates. It forms a long-grouped rhizoid, which arose from the stolon opposite rhizoids. This morphology is different from species in the Eurotiomycetes (Ascomycota) class, such as *Aspergillus* and *Ascosphaera*, which represent common subclasses of fungi that cause disease in honeybee larvae by germinating within the gut and ultimately mummifying the larvae when they ingest the fungal spores [47]. Furthermore, molecular analysis, based on the generated sequence of the *Rhizopus* 18S ITS region, confirmed that the fungus is *R. oryzae*. Also, BLAST analysis of the ITS region revealed a 99.83% sequence similarity with previously sequenced strains of *R. oryzae* [61]. Phylogenetic analysis also indicates that the isolated fungus in this study can be considered as *R. oryzae*.

### 4.2. The Virulence of R. oryzae for Honeybee Larvae Is Time- and Dosage-Dependent

*Rhizopus* is commonly detected in pollen, bee bread, and bee bodies. Early in 1974, *Rhizopus arrhizus* was isolated from the foraging honeybee midgut [49]. In 1988, *Rhizopus* sp. was identified in bee bread and the nurse honeybee midgut as disease-preventing fungi [51]. Recently, lower numbers of *Rhizopus* have been reported to be present in honeybee bread [50] and in stingless bee *Trigona collina* collection [62]. There have been many cases where bees collect fungal spores in place of pollen for unknown reasons [63,64]. Most collected fungi belong to either rust fungi, powdery mildews, or molds. Some of them may act as a form of nutrition or may be beneficial for bee health [63], while others have been found to spoil some soil-nesting bee bread and are a cause of larva death [48], including *Rhizopus*, whereas the mold-caused mortality of honeybee larvae is rare, and there are no data on the evaluation of the pathogenicity of these molds for honeybee larvae. The present study successfully isolated the strain of *R. oryzae* from brood mummy samples, and first evaluated its virulence for honeybee larvae. Our results revealed that the virulence of this pathogen for honeybee larvae was time- and dosage-dependent. This result is in line with the reports that *R. oryzae* showed pathogenicity for the soil-nesting bee, silkworm, wax moth [65,66,67], and fly [68]. It caused mortality at a lower concentration of 6.5 × 10^2^ CFU per larva g^−1^ in silkworms [66]. These documents reveal that *R. oryzae* can infect insects; here, we found that *R. oryzae* being virulent for honeybee larvae is not accidental. Previous research has found that *R. oryzae* causes disease in bee-pollinated plants [41,69]. Therefore, we hypothesize that *R. oryzae* may infect both the host plant and its pollinators after their visitation.

### 4.3. The Potential Influence of R. oryzae on Honeybee Colonies Requires Further Investigation

Although *Rhizopus* has been commonly found in honeybee bread and assessed causally for some bee diseases, this study is the first to report the detection of *R. oryzae* in managed honeybee colonies and an evaluation of its pathogenicity for honeybee brood in the laboratory. It may probably be due to the fact that *R. oryzae* is an opportunistic, facultative pathogen, like *A. apis*, which needs a specific trigger to switch from a saprobe to a parasite. In this study, *R. oryzae* was isolated from the samples of *A. apis*-infected bee larvae collected from four apiaries in China during the summer. When chalkbrood disease is prevailing, the weather is moist, and the state of disease of the colony is weak, this provides an ideal environment for *R. oryzae* growth. Additionally, due to their similar symptoms, the presence of *R. oryzae* may have been overlooked by beekeepers and researchers in the past. Thirdly, when co-existing in the hive, the dominant growth of *A. apis* inhibits the growth of *R. oryzae.* It can be speculated from the single-spore isolated method used in this study, that when the suppression by *A. apis* was absent, *R. oryzae* grew aggressively on PDA within 24 h. This result is consistent with the hypothesis that *A. apis* prevents other fungal growth when a mixed infection is present in the honeybee colony [15].

A honeybee colony is a superorganism; it possesses various potential defenses against microbial infections, such as propolis [70,71], hygiene behaviors [72], grooming behaviors [73], and colony thermoregulation [74]. A hygienic colony can detect infected brood earlier and will remove them from the colony before they become mummies, thus reducing the risk of disease epidemics [75]. There are several studies showing that honeybees can sense the spores of fungi and remove them by grooming. Taken together, although we detected the strong pathogenicity of *R. oryzae* for *A. mellifera* larvae, it did not raise the possibility of colony prevalence.

The complex interactions between disease-causing agents, such as fungi, parasitic mites, bacteria, and viruses, and environmental factors, such as plant protection agents and pesticides, can cause honeybee colony collapse [76]. Therefore, future studies are needed to determine whether this strain of *R. oryzae* has pathogenicity for honeybees at the colony level, and if it can interact with other causative agents or if it has undergone genetic mutations that may increase its potential to infect honeybee larvae under natural conditions. Even though *A. apis* has been the predominantly isolated fungus of honeybee larvae, as indicated by chalky mummies following pupation [77,78], our finding regarding *R. oryzae* indicates that this new fungal species may cause economically important disease levels in managed honeybee colonies, which can result in significant brood loss. Additionally, a comparative study at the genomic and transcriptomic levels of this strain and the *R. oryzae* strains found on plants may show whether a host shift from plants to insects has occurred. Further research is also needed to determine whether adult worker bees spread this pathogen and, if so, to identify the mechanisms and pathways involved.

## 5. Conclusions

To our knowledge, this is the first mycological and molecular identification of *R. oryzae*, which could cause disease in honeybee broods in China and may add to the existing challenges facing the honeybee industry. In this study, we successfully isolated the fungus, *R. oryzae*, from dead honeybee broods, and described the morphological characteristics of its hyphae, sporangiospores, sporangia, and sporangiophores. The taxonomic classification was confirmed, based on morphology and molecular identification. Koch’s postulates were used to verify the pathogenicity of this newly isolated fungus. Its virulence to honeybee larvae is both time- and dosage-dependent. In the present study, we hypothesize that host plant pathogens can be transferred to their pollinators, but the spread pathway and the pathogenicity for honeybee colonies need further investigation.

## Figures and Tables

**Figure 1 microorganisms-12-00313-f001:**
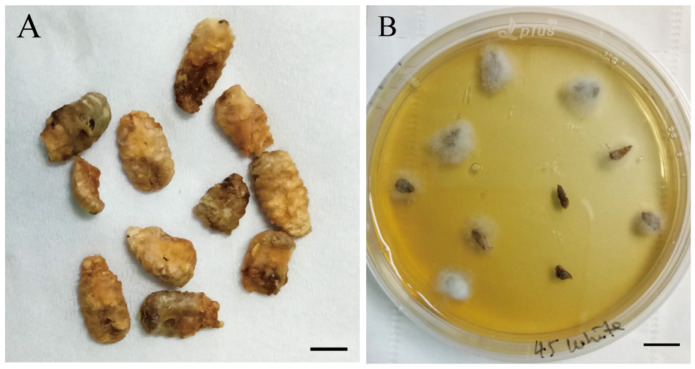
Collection of honeybee brood mummies and the selective inoculation of *R. oryzae* on PDA culture. (**A**) White mummies. (**B**) The growth of white *R. oryzae* mycelia from mummies on PDA culture. Bars = 1 cm.

**Figure 2 microorganisms-12-00313-f002:**
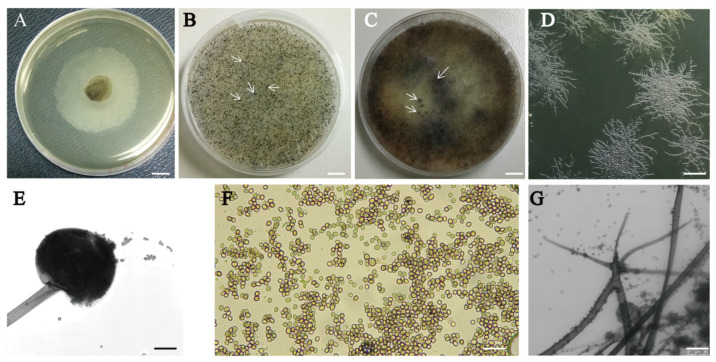
Mycelial growth pattern, venation, sporulation, and conidial structure of *R. oryzae*. (**A**) Mycelial structure. (**B**) Young conidia. (**C**) Matured conidia. (**D**) Mycelial venation. (**E**) Matured conidia head, observed with a 10× objective lens. (**F**) Conidial structure, observed with a 20× objective lens. (**G**) Mycelial rhizoid. The bars on A to C represent 1 cm, and those on D to G represent 50 μm. The white arrow points to the conidia.

**Figure 3 microorganisms-12-00313-f003:**
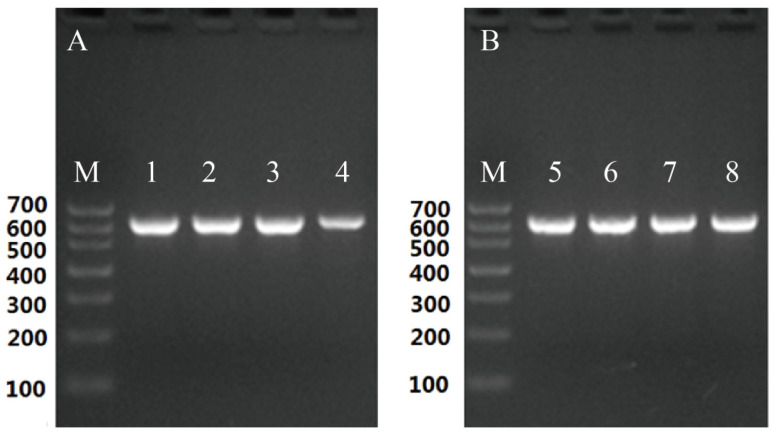
Internal transcribed spacer (ITS) bands of *R. oryzae* before and after honeybee worker larvae inoculation. (**A**) ITS of the isolated *R. oryzae* before the infection bioassay, where images 1–4 represent samples isolated from four beekeeping sites in China. (**B**) ITS of *R. oryzae* after infection bioassay, where images 5–8 represent samples isolated from four locations, respectively.

**Figure 4 microorganisms-12-00313-f004:**
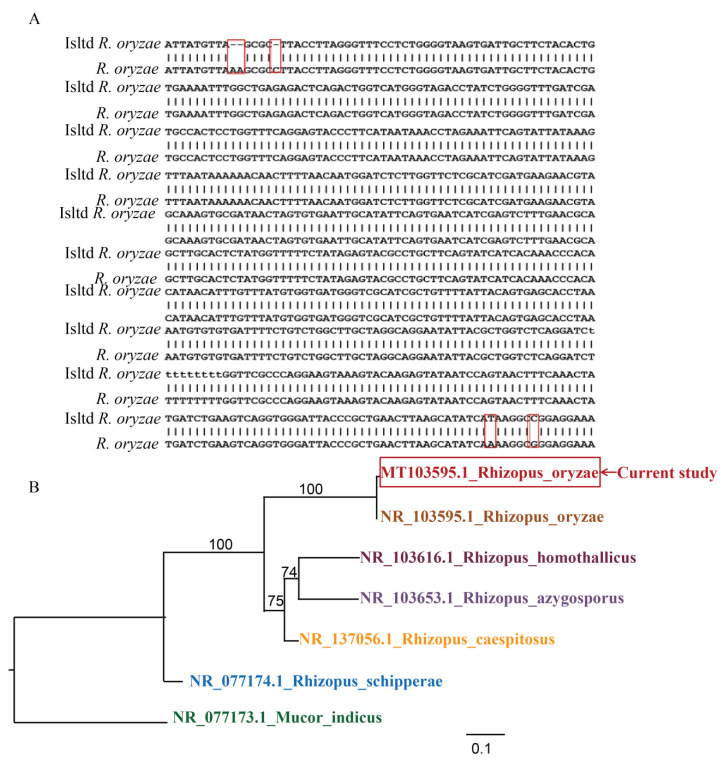
Complete sequence alignment and phylogenetic analysis of the ITS region of the isolated fungus. (**A**) Complete sequence alignment of the ITS region from the isolated *R. oryzae* to the NCBI-uploaded reference genome. Isltd *R. oryzae* represents isolated *R. oryzae*, *R. oryzae* represents the NCBI-referenced *R. oryzae*. The boxes show the base pairs of ITS with differences. (**B**) Phylogenetic analysis of the isolated fungus, based on the ITS sequence region. GenBank accession numbers for the sequences are adjacent to the corresponding species names. ‘MT103595.1_Rhizopus_oryzae’ in the red box represents the current sequence data from this study.

**Figure 5 microorganisms-12-00313-f005:**
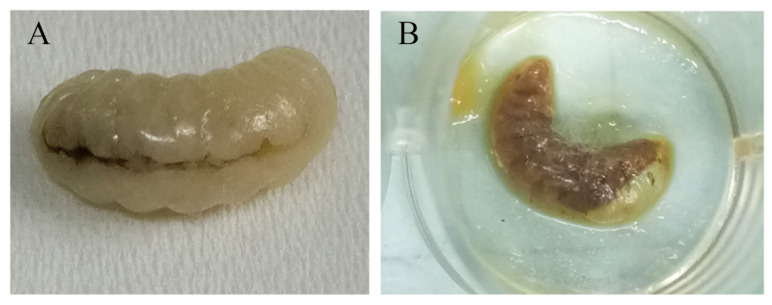
Observed physical expressions of *R. oryzae* inoculation in honeybee larvae. (**A**) Back apical midrib melanization in post-inoculation larvae. (**B**) Body melanization in post-inoculation larvae.

**Figure 6 microorganisms-12-00313-f006:**
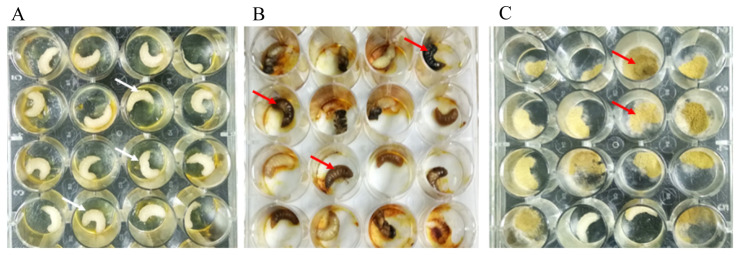
Observed physical expressions of *R. oryzae* on inoculated honeybee larvae compared with control group. (**A**) Non-infected control larvae. (**B**) *R. oryzae*: initially infected larvae. (**C**) *R. oryzae*: post-infection larvae. White arrows show health larvae and red arrows show infected larvae respectively.

**Figure 7 microorganisms-12-00313-f007:**
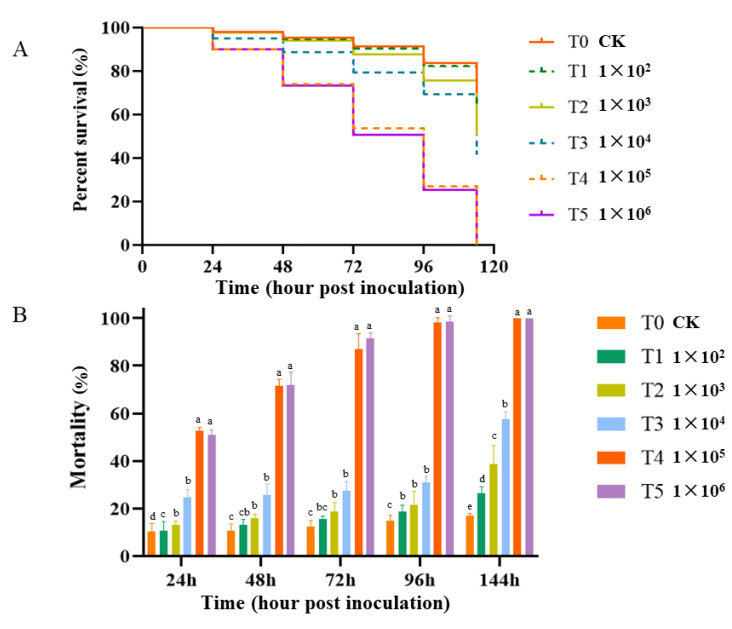
In vitro pathogenicity of *R. oryzae* and the mortality rate of honeybee worker larvae. (**A**) Cumulative mortality rate of larvae across inoculation times. (**B**) Percentage mortality rate of honeybee worker larvae at different conidial concentrations of *R. oryzae* and time. The error bars represent the SD. Lowercase letters show significant differences in the average larvae death rate among the concentrations. T1, T2, T3, T4, and T5 stand for *R. oryzae* spore-inoculated larvae at 1 × 10^2^, 1 × 10^3^, 1 × 10^4^, 1 × 10^5^, and 1 × 10^6^ conidia/larva, respectively. T0 is the control group, in which larvae were fed on a normal diet.

## Data Availability

The isolated fungus complete sequence has been uploaded to the NCBI; the GenBank accession number is MT1805651.1.

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
