# Peer review of "A New Isolated Fungus and Its Pathogenicity for Apis mellifera Brood in China"

_microorganisms, 2024, doi:10.3390/microorganisms12020313_

Round 1

Reviewer 1 Report

Comments and Suggestions for Authors

The paper is well comprehensible and brings clear message that pathogenic phytofungi may infect their associated pollinators. 

I think that hypertext links in chapters 2.2. and 2.3 do not reffer to specific methodology, but only to general web presentations, so I propose to remove it.

I have no fundamental objections to the article.

Reviewer 2 Report

Comments and Suggestions for Authors

The orientation in the Figure 7 could be easier for reader, if meaning of abbreviations T0-T5 would be repeated near the picture, not in text only.

It is very clear from the work that the fungus Rhisopus is pathogenic to the individual larva. What is the danger to the colony as a superorganism? We recommend expanding the discussion on this topic.

Reviewer 3 Report

Comments and Suggestions for Authors

Reviewer comments

Manuscript: microorganisms-2709044 - A new isolated fungus and its pathogenicity to brood of Apis mellifera in China.

The authors aimed to observe the occurrence of a new strain of the virulent fungus, Rhizopus oryzae, in cadavers from managed honeybee colonies in China. It also aimed to analyze the in vitro infection and pathogenicity of this fungus on honeybee brood. The species and strain specificity of the isolated R. oryzae from honeybees were confirmed through morphological observation and analysis of internal transcribed spacers (ITS) genes. The cause-and-effect pathogenicity of this fungal isolate was verified using Koch's postulates. Additionally, the in vitro pathogenicity of the pathogen on honeybees was tested by artificially inoculating worker larvae in an incubator. Probit regression analyses showed that the LT50 value against larvae was 26.8 hours at a conidial concentration of 1×10^5 conidia/larva, and the LC50 was 6.2×10^3 conidia/larva. This research on the new isolate of R. oryzae from honeybees suggests that it has significant pathogenicity to honeybee larvae and raises the possibility that pathogenic phytofungi may infect their associated pollinators.

The data analysis methods are correct.

The English of the text is well written and well readable but needs additional checking with a professional translator.

The uniqueness of the text is more than 95% by AntiPlagiarism.NET.

The text contains some misspellings and typos. Also need to expand the part of the discussion.

There are some comments and questions:

Lines 15, 208, 209 - ITS gene - should be - ITS region. - ITS is not gene.

Lines 40, 297 - Alought - should be - Although.

Line 45 - mytcooxins - should be - mycotoxins.

Line 47 - Mucor heinalis - should be - Mucor hiemalis.

Lines 52, 68 - entomo-pathogenic - should be - entomopathogenic.

Line 69 - R. Stolonifer - should be - R. stolonifer.

Line 73 - R. stolonifera - should be - R. stolonifer.

Line 84 - aikali bee - should be - alkali bee.

Line 111 - Qiangen - should be - Qiagen.

Line 112 - Theconcentration - should be - The concentration.

Line 191 - mcelial - should be - mycelial.

Line 230 - inoculateion i honeybee - should be - inoculation in honeybee. 

Line 247 - larvaeat - should be - larvae at.

Line 329 - MLwrote - should be - ML wrote.

In References - dio - should be - doi.

Please improve the manuscript according to the above comments.

Comments on the Quality of English Language

Minor editing of English language required.
